# Power Line Extraction and Reconstruction Methods from Laser Scanning Data: A Literature Review

Nosheen Munir * , Mohammad Awrangjeb and Bela Stantic

Institute for Integrated and Intelligent Systems, Griffith University, Nathan, QLD 4111, Australia
* Correspondence: nosheen.munir@griffithuni.edu.au

**Abstract:** Electricity has become an indispensable source of energy, and power lines play a crucial role in the functioning of modern societies. It is essential to inspect power lines promptly and precisely in order to ensure the safe and secure delivery of electricity. In steep and mountainous terrain, traditional surveying methods cannot inspect power lines precisely due to their nature. Remote sensing platforms, such as satellite and aerial images, thermal images, and light detection and ranging (LiDAR) points, were utilised for the detection and inspection of power lines. Nevertheless, with the advancements of remote sensing technologies, in recent years, LiDAR surveying has been favoured for power line corridor (PLC) inspection due to active and weather-independent nature of laser scanning. Laser ranging data and the precise location of the LiDAR can be used to generate a three-dimensional (3D) image of the PLC. The resulting 3D point cloud enables accurate extraction of power lines and measurement of their distances from the forest below. In the literature, there have been many proposals for power line extraction and reconstruction for PLC modelling. This article examines the pros and cons of each domain method, providing researchers involved in three-dimensional modelling of power lines using innovative LiDAR scanning systems with useful guidelines. To achieve these objectives, research papers were analysed, focusing primarily on geoscience-related journals and conferences for the extraction and reconstruction of power lines. There has been a growing interest in examining the extraction and reconstruction of power line spans with single and multi-conductor configurations using different image and point-based techniques. Our study provides a comprehensive overview of the methodologies offered by various approaches using laser scanning data from the perspective of power line extraction applications, as well as to discuss the benefits and drawbacks of each approach. The comparison revealed that, despite the tremendous potential of aerial and mobile laser scanning systems, human intervention and post-processing actions are still required to achieve the desired results. In addition, the majority of the methods have been evaluated on the small datasets, and very few methods have been focused on multi-conductor extraction and reconstruction for power lines modelling. These barriers hinder the automated extraction and reconstruction of power line using LiDAR data and point to unexplored areas for further research and serve as useful guidelines for future research directions. Several promising directions for future LiDAR experiments using deep learning methods are outlined in the hope that they will pave the way for applications of PLC modelling and assessment at a finer scale and on a larger scale.

**Keywords:** power lines; LiDAR; span; conductor; extraction

## 1. Introduction

Electricity is essential to the functioning of contemporary societies. To ensure the uninterrupted distribution of electricity, power lines must be effectively monitored and maintained. Transmission networks, regional networks, and distribution networks are typical components of electrical networks. One of the most important parts of the power transmission system is the high-voltage power line, which makes it possible to send electricity over long distances with little power loss [1]. The transmission system network is

expanding as a result of population growth and increased reliance on electricity. Due to the rapid expansion of transmission networks, it is therefore impossible to avoid mountainous terrains and forests within a power line corridor (PLC) [2]. The global length of high-voltage power lines has increased from 5.5 million kilometres in 2014 to 6.8 million kilometres in 2020 [3]. In the long-term, power lines are frequently obstructed by severe weather conditions (e.g., high temperature differences, humidity, and vegetation encroachment [4,5], which can amplify flash over discharge leading to large area blackouts, resulting in substantial financial costs and heavy national economic losses [6,7]. In order to ensure the safe and secure delivery of electricity, it is crucial to inspect the power lines promptly and precisely [8].

Traditional power line inspections are carried out through the use of field surveys. The inspectors keep an eye on the power lines and make estimation about the distance between them and the forest floor [9,10]. Due to the nature of this approach, the inspection cycle is lengthy and the workload is heavy. Some power line segments cannot be inspected on a regular basis due to the steep terrain and dense forest that surround them. On the other hand, helicopter methods have a lower detection rate because of the high speeds used and the crew's inability to simultaneously observe all possible problem types. Both methods, however, are reliant on the observations of the human eye. The use of video recordings and various cameras in addition to visual inspections is now commonplace. It is necessary to cover a large area when conducting power line surveys, and remote sensing techniques offer interesting alternatives.

Recent developments in hardware and data processing techniques have advanced remote sensing technology. There are various modern remote sensing methods (optical sensors [11,12], synthetic aperture radars (SAR) [13], thermal imaging [14], mobile laser scanning (MLS) [15], and light detection and ranging (LiDAR) [1,7,16] as well as several other monitoring devices (such as satellite, airborne, and unmanned aerial vehicles (UAVs) [17]. Each of these techniques has its own advantages and disadvantages in terms of PLC mapping. Optical satellite and aerial images, for instance, operate near infra-red wavelengths and passively gather energy from the earth's surface; these images are unsuitable for PLC mapping due to their passive mode; images cannot be captured in the dark and cannot extract or model power lines due to their thinness and proximity; however, they can detect wires [18]. In thermal imaging, sensors detect the infra-red radiation emitted by an object and create an image based on this data. Thermal imaging is not typically used for power line mapping and extracting PLC objects, but it is useful for detecting wire faults. SAR sensors take microwave images that can be taken in clouds as well but geometrical deformations and multipath scattering make SAR image analysis problematic for intricate power line structures.

Laser scanning technology captures and measures objects in the environment using laser beams. The 3D data generated by laser scanning are also referred to as a LiDAR point cloud. Scanning with lasers has come a long way in the last few years. Laser scanning systems, in contrast to conventional camera sensors, are not affected by lighting conditions because LiDAR can operate in the absence of external illumination [19]. Laser ranging information and the precise location of the LiDAR point can be used to create a 3D view of PLC [20], which is an excellent starting point for automated analysis. The resulting 3D point cloud makes it possible to accurately extract power lines and measure the distances between them and the forest beneath [21–23]. A point cloud created by LiDAR can be extremely dense and contains a wealth of information, thanks to its high precision, quick scanning speed, and ability to gather a large amount of spatial data.

A comprehensive review on various PLC surveying methods is given in Matikainen et al. [15], and it was concluded that airborne and land-based systems provide more detailed data than satellite images and are the most practical remote sensing method for extracting power lines. Numerous publications have focused on automated classification and reconstruction techniques for the extraction of power lines, specifically conductors. A typical procedure involves the generation of a digital terrain model (DTM), classification of the laser data

to distinguish power line points from other objects, especially vegetation, and 3D modelling of the individual conductors. Previously, emphasis was placed on classifying power line points using supervised and unsupervised ALS methods; in addition, the methods were tested on very small datasets or without visual or quantitative results [24,25]. With the advancement of ALS technologies, however, new methods are focusing on robust approaches for single and multi-conductor extraction and have been tested on complex terrains with large data sets [26,27]. In addition, with the advent of mobile laser Scanning MLS and UAV, numerous researchers have proposed new extraction and reconstruction techniques for power lines utilising MLS and UAV platforms [28,29]. The purpose of this review is to provide a critical review of the state-of-the-art laser scanning-based techniques for the extraction and reconstruction of overhead power lines. The review will provide an overview of the techniques and discuss the advantages and disadvantages of each technique.

Nearly 35 studies are included in total, the majority of which were published in scientific journals and conference proceedings. Following the overarching objective stated previously, our focus will be on discussing various applications in which the proposed algorithms have been used, as well as their applicability in real-world situations. The fundamental concepts of analysis methods are presented, along with some representative quantitative analysis results, in order to demonstrate the capabilities of various remote sensing techniques.

Due to the broad scope of the subject and the volume of research articles, it was not possible to include all studies or details of analysis methods. To begin, the data collection methods used in the PLC survey using LiDAR technology have been discussed in Section 2. For each type of data, some fundamental principles are introduced first, with an emphasis on aspects pertinent to power line surveys. This enables us to discuss the general merits and drawbacks of various approaches. Then, studies in the literature pertaining to power line mapping are reviewed and discussed in light of the findings of previous studies on power line extraction and reconstruction in Section 3. Finally, a summary including the future possibilities is presented in Section 4.1 and concluding remarks in Section 5.

## 2. LiDAR Data Collection Systems

Figure 1 shows categories of LiDAR data collection. In general, laser scanning systems are divided into two main categories:airborne- and terrain/ground-based systems [30]. Airborne acquisition allows us to cover large areas quickly. A terrain or ground-based acquisition system employs a LiDAR mounted on a vehicle or on a device on the ground (see Figure 2).

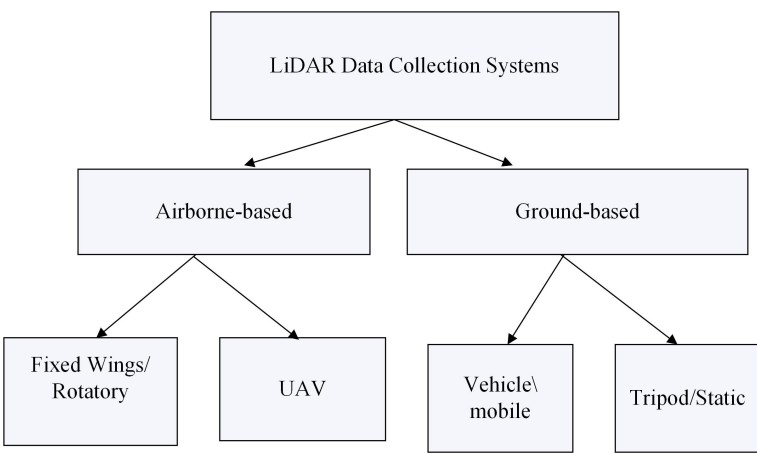

**Figure 1.** Different LiDAR data collection systems.

## 2.1. Basic Principles of Airborne- and Ground Based Laser Scanning

ALS uses LiDAR measurements from an aircraft (fixed-wings, rotary wings, UAV) to acquire environmental data (see Figure 2). The GNSS (Global Navigation Satellite System) and IMU (Inertial Measurement Unit) measurements reveal the sensor's orientation. The sensor generates data in the form of point clouds with coordinates (*x*, *y*, *z*). Additionally, the location of the pulses as well as their intensity are recorded. When a laser ray is pointed at an object, the receiver can detect up to four echoes per pulse [31]. Targets produce different echo signals for the same ray. Multiple echoes can represent detect treetops, intermediate plants, the ground, and other objects. The number of returning echoes and height discrepancies can be used to analyse the data. Full-waveform laser scanning records the shape of returning echoes.

In recent years, rapid technological advances in UAV component and battery technologies have increased the viability of UAV-based data collection for power line inspection. The increased availability of lightweight sensor systems and the development of UAV-related legislation have contributed to UAVs' commercial applicability.

Terrain laser scanning used ground-based remote sensing systems. These systems can be mounted on static tripods (Terrestrial Laser Scanning, TLS) or on vehicles (Mobile Laser Scanning, MLS). TLS scans closer to targets than ALS systems, allowing for collecting data with greater precision. In spite of this, MLS must travel on roads that do not always exist because of overhead high-voltage power lines.

The basic working principle of TLS and MLS is same as that of ALS, in In ground-based systems (TLS and MLS), the INS and the GNSS are mounted on vehicles or on tripods. These systems can move in all directions, including upwards. Once scans of a single area are complete, the tripod or vehicle is moved to another location to scan from another angle or capture data from a new area.

Stationary systems can produce more detailed and high-density point clouds because the sensor is kept perfectly still throughout the scan. This reduces the risk of point cloud outliers. Static systems can also move so multiple scans can be carried out of a single area from various angles, creating a more accurate and detailed picture of the environment.

## 2.2. Mapping of Power Lines from Airborne and Ground-Based Laser Scanning

The ALS data over power lines are typically collected by helicopter or by fixed-wing air planes, and the point density is typically in the tens of points per m$^2$ but can reach hundreds. On hard surfaces, the points have an absolute accuracy of 5–10 cm in the horizontal plane and 2–5 cm in the vertical direction [32]. Such information can be used to precisely map the shape of power lines. Because of the density and accuracy of the points, detailed mapping and monitoring of power lines and their surroundings are possible. Commercial companies have been mapping PLCs with LiDAR since 1995, and FLI-MAP has recently mapped thousands of kilometres of power lines [31,33]. A complete work flow of processing, analysis, and corridor clearance can be completed in less than 72 h as a state-of-the-art data acquisition method, allowing an airborne LiDAR data processing platform to provide uninterrupted rapid mapping services [34]. Helicopters have been used most frequently in power line studies because they can easily follow PLCs and take measurements from extremely low altitudes.

The use of UAVs carrying new sensors for electric power inspection has been developed largely since the end of the 20th century [35]. Given that the high-voltage power line facilities in remote or harsh environments are generally difficult to reach, UAV mapping presents huge advantages by saving manpower and producing more reliable results. UAV systems have been increasingly used by communities due to their low cost, less strict requirements for take-off and landing, and their ability to load different types of sensors (e.g., camera and LiDAR) [36]. As a light-weight and close-range ALS technology, UAV LiDAR, usually equipped with middle-sized or small laser sensors, has been rapidly developed for transmission line inspection as a time-saving and cost-friendly solution. It can directly generate dense point clouds in 3D with a higher level of precision compared

to aerial optical imagery and video [37–39] because image-based techniques can produce noisy results from the stereo-matching stage [39]. According to comparative studies of power line monitoring, the accuracy of height estimation for poles and trees is clearly better when using UAV LiDAR data than aerial images [9,40]. Compared to robotic inspection [9], UAVs have higher flexibility and larger coverage.

UAV is likely to become a popular technique in the future because it increases data acquisition flexibility while decreasing costs. ALS sees power lines and their surroundings from above, whether from helicopters or UAVs, which is useful for mapping power lines and trees near them. However, it also means that vertical poles and small components are not always visible in the data. Tree cover can also reduce power line visibility, and weather conditions can limit the ability to fly over power lines [29].

A variety of UAV platforms have been used for power line surveys. Fixed-wing UAVs, in general, must fly higher and faster and are best suited for vegetation monitoring and rough inspection of long power lines, whereas helicopter and multi-rotor UAVs can acquire detailed images by hovering close to the objects [17,41]. Deng et al. [41] proposed a multi-platform system made up of different types of UAVs designed for specific missions.

Ground-based laser scanning is a relatively new technology that has been primarily used to precisely map urban and street environments. It is particularly well-suited for corridor-type applications (for example, roads), and power lines are well-defined corridor-type objects of interest to some extent. MLS and TLS enable simple deployment and data collection with high precision and detail for asset modelling and monitoring. The mapping is not limited to the power line; it can also generate accurate data on the terrain and vegetation adjacent to the line. Outside of the corridor's roads, mobility can be accomplished by deploying an all-terrain vehicle (ATV) or a backpack-mounted LiDAR mapping system, also known as personal laser scanning (PLS) [42].

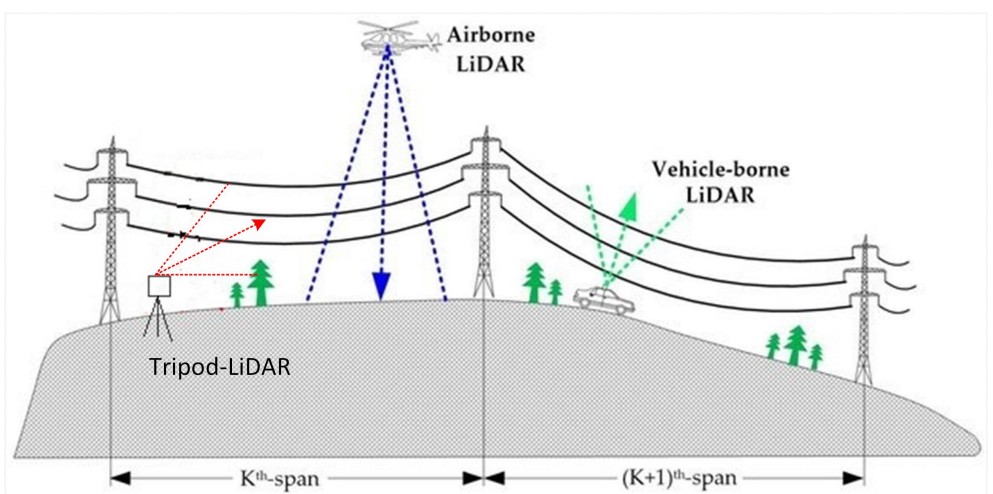

**Figure 2.** Power lines' mapping systems [43].

### 2.3. Advantages and Disadvantages of ALS, MLS and TLS in Power Line Mapping

The above-mentioned laser scanning systems types have their own advantages and disadvantages depending upon the application as shown in Table 1. They are further described in detail below.

**Table 1.** Comparison of LiDAR systems mounted on different platforms

| Platforms | System Abbreviation | Scanning Perspective | Density | PL Mapping Application | Weakness and Strength |
|---|---|---|---|---|---|
| Airborne | ALS | Top view | low | Forest mountain terrains | cost-effective and time-saving for large areas |
| Airborne | UAV | Top view | high | Forest mountain | Light weight, time-saving |
| Mobile/vehicle | MLS | side view | high | Urban areas, road side | cost-effective for small areas |
| Tripod/static | TLS | side view | high | Urban areas | cost-effective for small areas |

The primary advantages of ground-based surveys over ALS surveys are the increased detail obtained from close proximity to the objects of interest and the ability to operate in inclement weather (e.g., wind). For example, MLS operates on the vehicle, brings it much closer to power lines than ALS (0–20 m in MLS vs. typically tens of metres in ALS). Thus, both the point density of TLS (up to 1000 points per m$^2$ at a 10-m range from the scanner) and the three-dimensional precision of MLS (2–3 cm in good GNSS conditions (see [44–46]) are significantly greater than those of the ALS data. Increased point density enables a more thorough mapping of the surrounding environment, while increased point cloud accuracy enables the retrieval of more precise features such as surface normals and pole or trunk diameters. Additionally, the MLS/TLS laser beam is smaller in size (i.e., footprint) than the ALS laser beam, allowing for more detailed mapping of pylons and assets. As a result, TLS and MLS produce a more precise map of the power line components and corridor than ALS does. MLS and TLS scanning geometry, in comparison to ALS's, is better suited to mapping the components of power lines [15]. Because the ALS data are scanned vertically, it is difficult to detect power lines stacked on top of one another. From above, the uppermost power line casts a shadow on the ones below.

However, such power lines are easily visible in MLS data [21], as it is straightforward to control the sensor's trajectory to collect complete data on the power line components. Cameras are frequently included in MLS systems, and image data can be used to model and monitor the condition of power line components such as insulators [47]. The MLS is an excellent choice for collecting up-to-date information about power lines and potential damage, which would be extremely valuable following a natural disaster such as a storm. The effort required to collect data for an MLS campaign is significantly less than that required to shoot an ALS campaign and flying in restricted areas and in moderate wind poses no problems. When a LiDAR system is mounted on a tripod to collect data statically, it suffers from two major drawbacks: (1) point density decreases as the proximity from the scanner increases, and (2) occlusion occurs when other objects are present in between.

The use of MLS to map power lines in urban areas is justified, given the difficulty and inefficiency of flying a UAV or helicopter there. There have been few reports on the use of MLS in areas other than the road network, including forests. When mapping power line corridors, areas outside the road network differ in several ways from urban and road environments, including the roughness of the ground surface, mobility constraints (e.g., vegetation, rough terrain, and rocks) and satellite visibility (trees blocking signals). As a result, the positioning accuracy of the MLS system is reduced, and the point density along the trajectory in the point clouds becomes more heterogeneous outside of the road environment. Thus, extracting power line components automatically outside of a road network is more difficult than in an urban or road environment.

Acquiring an TLS system is less expensive, and a variety of LiDAR sensors are available to meet the grid operator's needs. Additionally, the operating costs are significantly lower than those associated with ALS. On the other hand, the TLS system cannot provide data covering a large area, particularly when the terrain is impassable to ATVs or other wheeled vehicles. Due to the lower acquisition and maintenance costs, multiple systems can be

purchased for the price of a single ALS system, allowing for the use of a more time- and space-efficient fleet for increased mileage. However, ALS systems are often cost-effective for locating and surveying high voltage transmission lines in wide and forest areas where personnel surveys and access by vehicles and humans are exceedingly difficult. However, for larger projects, the airborne LiDAR system could be more cost-effective. This is because an airborne LiDAR system can cover more ground in a shorter space of time than terrestrial LiDAR system—saving time and budget.

## 3. Processing Methods

Figure 3 shows how LiDAR data are processed for power line extraction. Processing issues with point clouds are typically brought on by their high data volume. Point clouds' high levels of detail make it difficult to properly extract power lines because they are packed with a variety of noise and other objects. To address this issue, methods available in literature first classify the power line scene to obtain power line points. Then, points for single conductors or multi-conductors are extracted to reconstruct them using mathematical models.

This section summarises numerous existing approaches that tend to remove undesired objects from the power line scenes to obtain the power line point class. They may begin with ground filtering, then execute supervised classification, or follow rules to identify power line points; however, the purpose is to obtain the power line point class. This step is an important prerequisite for the extraction and reconstruction of single or multiple conductor power lines.

Table 2 shows the summary of power line scene classification methods. The table details the classification procedures involved for extraction of power line points as a pre-processing step to remove unwanted object points or to extract power line points as one class. Typically, methods used supervised [48] or rules [49] to obtain power line points. The second column in Table 2 shows the classification type, followed by the method used for the given classification in the third column. The fourth column indicates whether ground filtering was applied prior to classification. The fifth and sixth columns describe the neighbourhood and features used for classification, followed by the density of evaluated data sets and the precision of the results in seventh and eight columns, respectively. Each of these columns of Table 2 is explained below in detail:

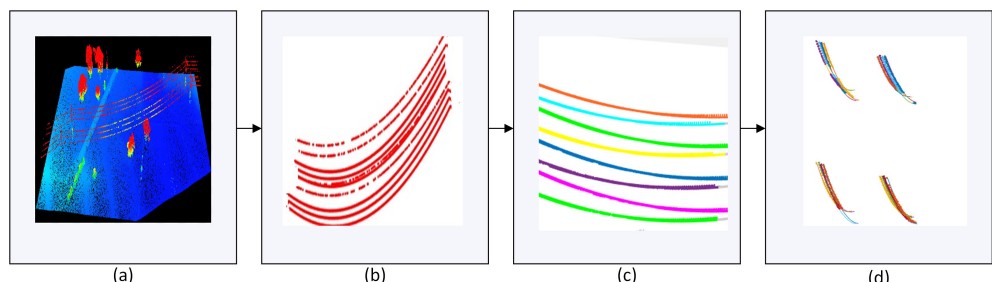

|   (a)   |   (b)   |   (c)   |   (d)   |

**Figure 3.** LiDAR data processing methods for power line mapping. (**a**) LiDAR point cloud data; (**b**) separation of power line points Section 3.1, Table 2; (**c**) single power line extraction and reconstruction Section 3.2, Table 3; (**d**) multi-conductor power line extraction and reconstruction, Section 3.2, Table 4.

### 3.1. Classification of Power Line Scene

3.1.1. Classification Type

The second column of Table 2 shows the classification type. The present methods for classifying power lines can be divided into supervised classification and rules-based/unsupervised classification depending on the approaches used to solve the classification problem.

**Table 2.** Power line scene classification. PCA—Principal component analysis; ML—Machine learning; HT—Hough transform; NA—Not available; TIN—Triangulated irregular network; CSF—Cloth simulation filtering; RTF—Recursive terrain filter; DTM Digital terrain model; DSM—Digital surface model; RF—Random forest; JB—Joint boost; SVM—Support vector machine; EC—Euclidean distance.

| Author | Method | | | | | Density | Results |
| | Type | Technique | Ground Filtering | Neighbourhood | Feature | (Pt/m$^2$) | Accuracy (%) |
|---|---|---|---|---|---|---|---|
| Jwa [50] | -Rules | -Segmentation -Filtering | × | -Voxel | -HT -Eigen -Density | 24 | NA |
| Munir [51] | -Rules | -Statistical analysis | -Height | -Voxel | -Vertical profile | 24 | NA |
| Huan [52] | -ML | -Object-based -RF | × | -Segment surface | -Eigen -Intensity | NA | 75.3 |
| Liang [25] | -Rules | -Software | × | NA | NA | 1.2 | NA |
| Yang [53] | -Rules | -Height filtering | TIN | NA | NA | 41–100 | NA |
| Melzer [54] | -Rules | -Culling filtering | -DTM | -2D grid | NA | 25.5 | NA |
| Otega [55] | -Rules | -Statistical -Image-based | × | -Pixel | -Height -Intensity -Return number | 307–402 | 97.4 |
| Zhang [29] | -Rules | -Statistical | -Height -DEM -DSM | -2D grid | -Density -Height | 307–402 | 96 |
| Guo [56] | -ML | -Point-based -JB | × | -Voxel -Sphere -Cylindrical | -Geometric -Radiometric -Topographic | NA | 88.69 |
| Zhou [57] | -Rules | -Segmentation | -Height | -2D grid -3D grid | -Height | 213.6 | 98.18 |
| Jung [28] | -Rules | -Height -Image filtering | -Morphological | -3D grid | -Eigen | 213.6 | NA |
| Lehtomaki [58] | -Rules | -PCA | -Software | -Voxel | -Eigen | NA | NA |
| Zhenwei [59] | -Rules | -PCA | -CSF | -K-d tree | -Eigen | NA | NA |
| Fan [60] | -Rules | -Filtering | -DTM -Height | -Voxel | NA | NA | NA |
| Gaha [61] | -Rules | -EC -RANSAC | -Morphological | -Cylindrical | NA | NA | NA |
| Yadav [19] | -Rules | -Segmentation | -Height | -2D grid | -Density | NA | NA |
| Cheng [21] | -Rules | -Filtering | -Height | -Voxel | -Density -Eigen -HT | 19 | NA |
| Guan [22] | -Rules | -Filtering | -TIN | -Voxel | -Density -size -shape | 286.4 | NA |
| zhang [62] | -ML | -Object-based -RF | × | -Segment surface | -Eigen -Intensity | 1.2 | 75.3 |
| Kim [48] | -ML | -Object-based -RF | -RTF | -Segment | -line | 30 | 98.5 |
| Kim [63] | -ML | -Point-based RF | -RTF | -Voxel -Sphere | - Geometric -Radiometric -Topographic | 30.29 | 98.5 |
| Wang [10] | -ML | -Point-based -SVM | Tiff toolbox | -Cylindrical - Sphere -Slant cylindrical | -Geometric distribution | 30 | 98.04 |

Supervised classification is the classification of PLC points into different classes using a machine learning classifier. Machine learning is a powerful supervised statistical method that can be used to separate power line points from 3D LiDAR data. Popular classifiers for laser point classification include Support Vector Machine (SVM) [10,62], Random Forests [48,64], JointBoost [56], and others. Thus, supervised classification necessitates training data, features, and a classifier.

The need for training data is the greatest drawback of supervised algorithms, yet they are excellent for multi-class classification. By utilising machine learning and automatic feature selection, a significant amount of the arduous design and programming work inherent in conventional voxel feature classification methods is eliminated. Consequently, for complicated landscapes due to their efficiency, machine learning classification techniques are preferred to traditional/rule-based classification techniques for classifying objects with many kinds.

Rule-based classification attempts to discover general rules that are based on piecewise linearity and other geometric characteristics of power lines. Usually, this approach focuses on specific object types. It can attempt to discover general rules that can be used to classify point clouds of many different kinds of landscapes, but a disadvantage is that it makes many assumptions. This classification approach does not require training data; hence, it is unsupervised. As rules are derived from the data sets, therefore the classification rules are not necessarily transferable from one point cloud to another.

### 3.1.2. Method

An approach based on supervised classification is typically used object-based [48] and/or point-based [63] methods for classification (see Column 3, Table 2). Segmentation-based classification methods aggregate related objects in segments using different algorithms such as surface-grown [62], region-grown [52], and extract features from each segment. Usually, this approach focuses on classification of specific object type and divides all point clouds into one geometric form, which is irrational. Despite the limitations, segment-based classification methods have two main benefits over point-based classification methods: (i) segments help compute geometric features, relieving the need for neighbourhood optimisation [56,65] and (ii) segments give several new attributes that can be used to apply semantic rules [52]. On the contrary, the point-based classification separates the point clouds into different classes directly. Segment-based classification methods produced low accuracy for wire class if planes are used to detect object which is not suitable for wires [52,62]. These methods are good for building classification but offer low accuracy for wires as wires are of line geometric shape. Kim et al. [48] achieved better accuracy as they remove the ground point beforehand and used lines to detect segments which are suitable for wires.

On the other hand, rule-based classification is on hierarchical rules to achieve classification [28,60]. These rules may include statistical analysis [50,51] or hierarchical filtering procedures [21,28].

### 3.1.3. Ground Filtering

The fourth column of Table 2 shows different methods to perform ground filtering prior to classification. Ground and height filtering is critical prerequisites for reducing unwanted objects. The point cloud data for power line extraction include not only power lines but also a large number of outliers that are lower than power lines (ground, shrubs, low buildings, and so on), which will impede extraction. Ground is considered the most dominant class in datasets, whether from the urban or forest areas. The elimination of the dominant class improves classification accuracy [48] and makes the following step more efficient [28]. Ground point classification is an important step in creating a DTM (Digital Terrain Model) from LiDAR point cloud.

Researchers have used a variety of LiDAR data ground filtering algorithms, such as morphological ground filtering [28,61], cloth simulation [60], triangulated irregular network (TIN) [22], and RTF (Recursive Terrain Filter) [63]. The TIN densification filtering algorithm is thought to be robust and stable for modelling discontinuous surfaces such as urban areas [66,67]. In addition, a simple morphological filter (SMRF) and a multi resolution hierarchical filter (MHF) rank first in urban and forest areas, respectively [68]. The cloth simulation filtering (CSF) algorithm has recently gained popularity on relatively

flat terrain. Because it has small number of parameters and simple to configure, most objects in the cropped data are lower than power lines.

Since there is typically vegetation below power lines, some methods use height criteria to remove LiDAR points below or above the lines. These height thresholds are 4 m [28], 6 m [22], and 5 m [60]. However, height filtering alone is ineffective because tall or long buildings or trees will not be eliminated. For example, points which are 4 m above the ground are the candidate points for power lines. However, If the terrain is complicated, it is impossible to eliminate ground points using the absolute height of the data points alone.

### 3.1.4. Neighbourhoods

For classification of power lines either by using supervised or unsupervised methods, methods make use of different 2D [19,55] and 3D [50] neighbourhoods to compute the properties of individual points for power line point classification (see forth column Table 2). The 2D neighbourhood either utilize a grid to interpolate the 3D point cloud into 2D grid as a raster image, and each grid (pixel) represents one object points [55], or the methods divide the point cloud onto 2D grid data [19]. The 2D neighbourhood provides the advantages of efficient management of enormous quantities of LiDAR. However, the grid-based neighbourhoods implicitly assume that each grid represents one object only; however, pixel represents only, vertical overlaps exist between wire and terrain, vegetation and wire, wire and pylon, so this the main limitation of the grid-based neighbourhood.

Determining the type and size of a neighbourhood is vital [63]. Methods select scale values using heuristic or empirical knowledge of the scene or data. Large neighbourhood sizes are efficient, but they may contain several object points, whereas small neighbourhood sizes may have few or no object points and require more time to estimate features. The smaller neighbourhood makes it more difficult to identify wires and other objects. Neighbourhoods can be spherical [56], cylindrical [64], voxel [63], or k-closest [10]. Wang et al. [10] introduced a new neighbour (slant cylindrical) based on power line direction. Mclaughlin [69] discussed the disadvantage of using ellipsoid neighbourhood in their paper, i.e., if neighbourhood has the intersection for vegetation and transmission line will lead to in classified data. If an ellipsoid grid has a few points in direction of power lines, they will be classified as vegetation. Features can be derived from one neighbourhood type with a variable scale parameter or from many neighbourhoods.

The sphere-based method was more exact and reliable and had a high success rate than the voxel-based method. Nevertheless, due to its quick feature extraction, voxel-based neighbourhood can rapidly categorise large-scale corridor data. Mostly, the methods voxel-based neighbour hood for rules-based classification (see Table 2). Wang et al. [10] compared the classification of power line scene and argued that spherical neighbourhood yielded superior results in comparison to voxel, cylindrical, and k-nearest neighbourhood, and also claimed that slant cylindrical provided the best performance. Determining the ideal neighbourhood size for each location is still challenging unfortunately, as these neighbourhood optimisation methods necessitate repeated calculations of eigenvectors and eigenvalues for each point; hence, they are somewhat time-consuming; this is the primary drawback of this type of classification [52]. In object-based supervised classification, the segment is considered as neighbourhood, and each segment represents geometric structure by using model-fitting-based methods or region-growing-based methods [52].

### 3.1.5. Feature

The proper identification of power line points is contingent on the extraction of valuable features that can differentiate power lines from other objects [10]. There are two primary kinds of features for identification power line points: reflectance-based or radiometric features and topographic or geometric features. These features can be estimated using 2D or 3D neighbourhood or directly from the segments. Frequently, radiometric parameters are associated with the intensity [25] and echo [55] captured by scanner systems. Therefore, the uniqueness of these types of characteristics is strongly dependent

on the signal quality of the scanner. The common sets of features that are used for a 2D neighbourhood are Hough transform (HT), height, intensity and return number. The groups for power line components have lower and less diverse returns than the rest of the categories [55]. Power lines, pylons or dense vegetation areas imply a low reflectivity and hence show low intensity values. However, power lines have very similar low intensity values to trees, which made it difficult to discriminate them from their intensity values [25]. The initial classification by using intensity, echo, and return by the first can include false positives and require post processing to improve classification accuracy [25,55].

The 3D features are computationally extensive but preserve the 3D details. The common sets of features in 3D neighbourhood are eigenvalues (power lines represent very large); HT to find the lines and density which is low for power line points in a given neighbourhood as compared to pylons and vegetation. The HT is an effective tool for straight line detection. However, the standard HT incurs a high computational cost and often results in the detection of spurious lines [25,55]. The computation of eigenvalue ratio is much faster than HT [50]. The contemporary methods for determining power line characteristics make extensive use of geometric features [52]. The 3D features provide greater precision than grid-based features [63]. Their characteristics are more suited for objects that overlap.

It is important to note here, for object-based supervised classification, that the features are extracted from segment characteristics. These characteristics contain qualities between objects. Segmentation-based features are helpful to compute geometric features, relieving the need for neighbourhood optimisation [56,65] methods, and segments give several new attributes that can be used to apply semantic rules [52]. However, this approach focuses on classification of specific object type.

### 3.1.6. Density and Classification Accuracy

The last column of Table 2 shows the classification accuracy. Most of the rule-based methods lack classification accuracy results. The majority of methods for rule-based classification use this technique as a pre-processing step for the extraction of single power lines. The elimination of ground points prior to classification improving the classification accuracy can be seen in Table 2 [10,63]. This is due to the fact that ground is a major class while pylons and wires are minor classes, so the imbalanced training will affect the classification accuracies. Kim et al. [64] observed that imbalanced learning reduced classification accuracy. It can also noted that the RF [63] and SVM [10] classifiers yielded better classification accuracies, i.e., 98% as compared to the JB classifier [56], i.e., 88%. In addition, Kim et al. [48] used segment as a line for feature extraction as compared to surface [52,62] and yielded better results. Overall, classification using multi-classifier achieved the highest classification accuracy [64]. Post-processing, such as contextual limitations, can also improve classification. Guo et al. [56] improved power line classification accuracy from 86.5% to 87.1% by adding graph-cut segmentation as a post-processing step.

### 3.2. Power Line Extraction and Reconstruction

After separating power line points from other object points such as vegetation, pylons, and buildings, the next step is to identify the points that belong to individual power lines. As mentioned in Section 3.1, the previous methods classified PLC points into various classes using both supervised and unsupervised methods. However, for reconstruction, the point cloud data have to be classified, and points on single power lines need to be rendered. The wires are very thin as compared to buildings, trees and pylons; thus, the actual number of points reflected from wires is far smaller than the number of input points. This issue makes single power line rendering very difficult [27]. In this category, the techniques for single and multi-conductor power line extraction and reconstruction have been reviewed. The proposed methods, algorithms, and results are compared, and their real-time application effectiveness is estimated.

**Table 3.** Single power line extraction using ALS. LS—Least square; PL—Power line; HT—Hough transform; CLF—Compass line filter; EC—Euclidean clustering; NM—Numerical method; SC Stochastic constraints; NA—Not available; $C_p$—Completeness; $C_r$—Correctness.

| Author | Method | Extraction | | Reconstruction | | Density | Results Extraction | | Results Reconstruction |
|---|---|---|---|---|---|---|---|---|---|
| | | Grid | Span | Curve | Technique | Pt/m$^2$ | Point-Based | Object-Based | Fitting Error |
| Mclaughlin [69] | -Piecewise -Mean and dominant Eigen | -Ellipsoid | ✓ | -Catenary | -NM | 2.5 | NA | $C_r$ 72.1 | NA |
| Jwa [50] | -Piecewise -CLF outlier testing | -Cuboid | ✓ | -Catenary | -SC | 24 | NA | -$C_p$ 96 | <0.05 m |
| sohn [70] | -Piecewise -CLF outlier testing | -Cuboid | ✓ | -Catenary | -SC | NA | NA | -$C_p$ 100 | <0.05 m |
| Guo [71] | -Profile segmentation -Geometry matching ratio | NA | ✓ | -Polynomial | -RANSAC | NA | NA | -$C_r$ 89.8 | NA |
| Munir [72] | -Profile segmentation -HT -Geometry matching ratio | NA | ✓ | -Polynomial | -LS | 23.6 to 56.4 | NA | -$C_p$ 97.6 -$C_r$ 99.5 | 0.0001 m |
| Liang [25] | -CLF -RANSAC -Linear model | NA | × | -Polynomial | -NM | 1.2 | $C_r$ 96.5 | NA | NA |
| Yang et al. [53] | -EC -RANSAC | 3D grid | × | -Catenary | -LS | 1.2 | NA | -$C_p$ 98.1 -$C_r$ 95.9 | NA |
| Melzer [54] | -Clustering -HT -Neural gas network | NA | ✓ | -Catenary | -RANSAC | NA | NA | NA | NA |
| Zhang [29] | -Clustering -Eigen -Height -Spatial continuity -CLF | -3D grid | ✓ | -Catenary | -RANSAC | 307–402 | -$C_p$ 96.3 -$C_r$ 96 | AN | 0.241 m |

Tables 3–5 summarise the methods used for single conductor and multi-conductor extraction using ALS and MLS data, respectively. All of these approaches use some methods such as clustering and geometry matching, segmentation to extract the points belonging to each power line in single or multi-conductor configuration (see Column 2 of Tables 3–5).

### 3.2.1. Method

The traditional single power line extraction methods are mainly based on region growing [50,69]. The algorithms begin with an arbitrarily chosen neighbourhood (ellipsoid or voxel) in the direction of power line, and then generate models by estimating parameter of members in group and iteratively add adjacent matched neighbourhoods with similar parameters, progressively refining the catenary parameters. However, the extraction and reconstruction of individual power lines could fail in some situations causing power line to be divided in two or more power lines or undetected. This is primarily due to sparseness of the data, resulting in a failure to first compute a reliable local model or to find the connected neighbourhood. These approaches are extremely dependent on the direction of the power line. Using campus line filter (CLF), Jwa and Sohn [50] approximated the power line direction of a classified power line points. Similarly, Mclughlin [69] emphasised in their work that the orientation of the power lines should be parallel to the major axis of the ellipsoid neighbourhood; otherwise, the approach will fail to extract power lines.

In order to overcome these challenges, several clustering techniques, such as Euclidean clustering [53] and agglomerative clustering [73], have been utilised to group individual power line points into clusters and power line clusters merged using spatial continuity and local collinearity [29,53,74]. However, clustering can work in ideal situations when the density of data is high, and there are moderate gaps between power lines. However, in practice, the extraction and reconstruction of power line could fail in certain situations, resulting in multiple clusters for the same power line because the algorithm is unable to find the merging clusters due to the gap [28,53,74].

Typically, prior to clustering, some approaches convert the classified data to a 2D grid [19] or an image [75], and then utilise HT [75] or RANSAC [58,61] to extract the lines from the classified power line points. In the grid size, the power line should be isolated within a horizontal segment. In addition, the resolution of the grid is determined by the sparsity of a given LiDAR point cloud. A low resolution will reduce the size of the image, making the individual objects on the power line appear too small, but a high resolution will make the points in image too sparse.

The HT algorithm is an effective tool for obtaining the straight line detection. However, the standard HT incurs a high computational cost and often results in the detection of spurious lines [25,25]. However, for the RANSAC algorithm, the computational complexity is highly sensitive to the user parameters. In both cases, if the two wires are located close to each other, it is difficult to separate them.

Some other approaches [71,76] used profile segmentation with a region growing algorithm to connect comparable points belonging to each individual line in a given span in order to circumvent the limitations of clustering and piecewise algorithms; however, they largely relied on pylon positions.

### 3.2.2. Grid Transformation

For extraction of single or multi-conductor power lines, some methods transform the point cloud into a 2D grid [19,69] or to a 3D voxel grid [28,51,77]. The 2D grid-based methods have limitations, as they interpolate the 3D point cloud into a 2D grid, which causes the overlapping of multiple objects [51,73]. Among these grids, a voxel grid is considered as more efficient and reliable [29]. The grid selection and size is an important factor for the extraction of single power line points. The 3D voxel grid helped to compute the geometric features of power line by ensuring that other objects were separated from voxel blocks containing power lines [29].

The size of grid depends on various factors such as sparseness of data [69], the cross section of bundle conductors [29], and span length [50]. To determine an adequate neighbourhood size, two parameters were considered: the distance between two power lines to avoid a situation in which the target points are mixed with other power line points; and the density of the community to provide sufficient points. Usually, the size of $5 \times 5 \times 5$ m$^3$ for voxel is used in many papers [29,51].

### 3.2.3. Sub-Conductor

Extraction of individual sub-conductors from bundles is generally an uncommon area of research; most existing methods extract power line points as a class or consider bundle conductors as a single conductor for power lines extraction and modelling [51]. Multi-conductor spans comes in different configurations, i.e., four sub-conductors in a bundle or two sub-conductors in a bundle. As shown in Figure 4, these conductors in a bundle are very close to each other, and the extraction of sub-conductors is not an easy task. Notably, the precise modelling of power line is highly reliant on the accurate extraction of each sub-conductor. Identification and extraction of bundle conductors are essential for accurate modelling and mapping of each power line. A few studies have concentrated on extraction and reconstruction of sub-conductors.

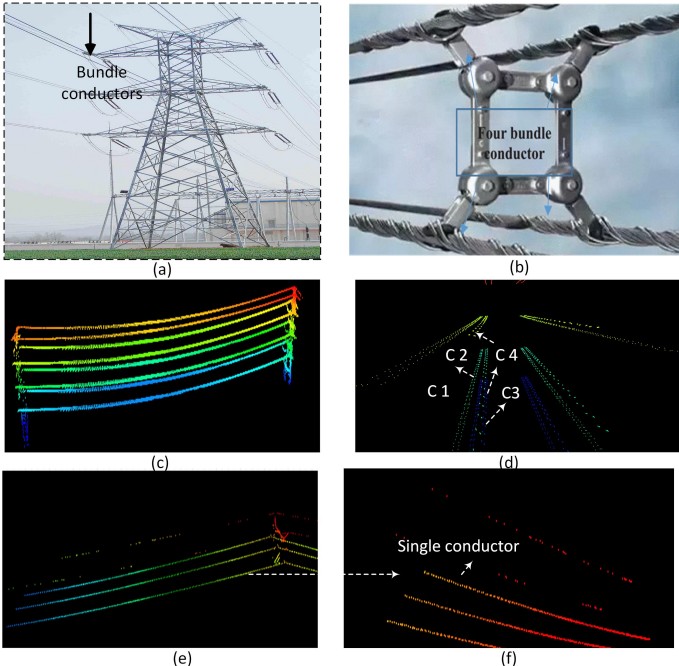

**Figure 4.** Sub-conductors configuration. (**a**) high-voltage transmission line tower with bundles in sub-conductors configuration; (**b**) magnified version of bundle from (**a**); (**c**) LiDAR data for span with bundles in four sub-conductors configuration; (**d**) magnified version of bundle from (**c**); (**e**) LiDAR data for span with single power line; (**f**) magnified version of bundle from (**e**).

These methods start with classification of the power lines scene using the method described in Section 3.1. Column 4 of Table 4 shows the methods which attempt to extract and reconstruct a sub-conductor. For sub-conductor extraction, most of the work conducted by Munir et al. [26,51,75,76] using both image-based and point-based methods. All techniques utilised ALS data with multi-bundle configurations on corridors that traverse outside of urban regions. These approaches have been evaluated on datasets with a very low point density. With excellent precision, Munir et al. [26] and Zohu [74] are able to extract up to four sub-conductors.

### 3.2.4. Span

A few approaches also indicated that pylon placement can facilitate the extraction of power lines in the form of spans [27,51]. These methods make use of the distribution characteristics of power lines, i.e., similarity, parallelism, identical sag and direction between neighbouring pylons. The additional information, e.g., power line distribution within a span and contextual pylon data, helps with the improvement of power line extraction and reconstruction. Locating spans relies heavily on pylon detection. These methods first detect the location of pylons in order to locate spans, or they use predefined locations from the dataset. This will reduce the size of the point cloud, making it easier to extract power lines in each span [71]. Additionally, it reduces the difficulty of analysing LiDAR data.

Extracting pylons prior to power lines is easier in forest settings because the size and height of pylons is much larger than that of trees, and pylons in forest locations are in predefined corridors [26], Nonetheless, if pole recognition had been performed prior to the removal of electricity lines in metropolitan areas, a significant proportion of tree trunks would have been misidentified as poles because of height similarities [58].

### 3.2.5. Reconstruction

After rendering individual power lines, they are using different fitting algorithm. Due to the fact that the power line model is nearly identical to the catenary and, with sagging posture, power lines are often modelled with a second order polynomial equation in 3D [27,78], catenary [29], and parabola [19]. As discussed above, some methods have carried out extraction and reconstruction together using a piece-wise technique [69,79]. Alternatively, power lines can be modelled separately with a straight line in the horizontal plane and catenary in the vertical plane by estimating parameters using numerical methods [73].

In literature, many mathematical models such as catenary curve, piecewise line, and parabola have mostly been used in conjunction with a RANSAC [29,54,71] or least square (LS) [53] algorithm for power line reconstruction. Many methods used CLF to find direction and then used RANSAC [78] orthogonal distance [29] to find the points that belong to each line. RANSAC [71,74] and least square [53] algorithms are also used for estimating parameters for the catenary curve. The RANSAC algorithm is more robust to the outliers [72], but the reconstruction model heavily relies on reliable initial parameter settings [71] and on effective seed section [71].

### 3.2.6. Results

Although there have been many advances in power line extraction from point clouds, notable limitations can be summarized in terms of efficiency, versatility, and robustness. Reliance on supplemental data, such as predefined pylon locations [71] and pre-classified data, limit the versatility of the approaches to a specific system (e.g., MLS, ALS, TLS)). Furthermore, many approaches have been tested and optimised on a limited number of datasets and, therefore, may not scale to work effectively with other datasets acquired in diverse conditions (e.g., urban, rural, and forest areas) and a reliable parameter setting is still an issue in many studies. It is impossible to draw a precise conclusion about achievable accuracy or compare various approaches due to the wide variety of study areas and quality evaluation methods. Some methods [26–28], have collected the ground truth points to estimate the point-based completeness, correctness and quality. While some other methods [19,50,61], have estimated the object-based accuracy by counting and comparing the number of power lines detected, undetected, and split.

Only a few visual demonstrations have been used to present the findings in numerous studies. Studies with larger tests conducted under realistic conditions and numerical quality analyses appear to be rare. Due to the slenderness of power lines, the laser point density must be relatively high for effective power line extraction. The extraction process can be complicated by vegetation and the complexity of the power line network. Because power lines are such narrow objects, the density of the laser points should be relatively

high in order to achieve good power line extraction results. The surrounding vegetation and the complexity of the power line network can make extraction more difficult.

Tables 4 and 5 show point-based and object-based power line extraction accuracies for these methods. Mostly, the methods which have extracted power lines in the form of spans have achieved high completeness and correctness [26,50] as compared to others which did not consider pylon locations [28,58].

**Table 4.** Power line multi-conductor extraction using ALS. LS—Least square; PL—Power line; HT—Hough transform; CLF-Compass line filter; EC—Euclidean clustering; NM—Numerical method; SC Stochastic constraints; NA—Not available; C$_p$—Completeness; C$_r$—Correctness.

| Author | Method | Extraction | | | Reconstruction | | Density | Results Extraction | | Results Reconstruction |
|---|---|---|---|---|---|---|---|---|---|---|
| | | Grid | Sub-Conductor | Span | Curve | Technique | Pt/m$^2$ | Point-Based | Object-Based | Fitting Error |
| Munir [72] | -HT -Profile segmentation | -NA | 4 | ✓ | -Catenary | -RANSAC -LS | 23.6–56.4 | NA | -C$_p$ 95.9 -C$_r$ 100 | |
| Munir [26] | -Image-based -Clustering | NA | 4 | ✓ | NA | NA | 23.6–56.4 | -C$_p$ 97.1 -C$_r$ 100 | -C$_p$ 92 -C$_r$ 95 | NA |
| Awrangjeb [27] | -Segmentation -Image-based | 2D plane | 2 | ✓ | -Polynomial | -NM | 23.4–56.4 | -C$_p$ 95 -C$_r$ 100 | -C$_p$ 92.6 -C$_r$ 99.6 | 0.012 m |
| Munir [76] | -EC -Line fitting | 2D plane | 4 | ✓ | NA | NA | 23.6–56.4 | -C$_p$ 99.15 -C$_r$ 100 | -C$_p$ 99 -C$_r$ 100 | NA |
| Munir [51] | -Point-based -Voxel alignment | -Voxel | 2 | ✓ | NA | NA | 23.6–56.4 | -C$_p$ 97.9 -C$_r$ 98.9 | -C$_p$ 92.5 -C$_r$ 96 | NA |
| Zohu [74] | -2D plane projection -Line fitting | NA | 4 | × | -Catenary | -RANSAC | | NA | -C$_p$ NA -C$_r$ 98.9 | 0.02 m |

**Table 5.** Power line extraction using MLS. LS—Least square; PL—Power line; C$_p$—Completeness C$_r$—Correctness; EC—Euclidean clustering (EC); Ht—hough transform; NA—Not available.

| Author | Method | Extraction | | Reconstruction | | Density | Results Extraction (%) | | Results Reconstruction (m) |
|---|---|---|---|---|---|---|---|---|---|
| | | Grid | Span | Curve | Technique | Pt/m$^2$ | Point-Based | Object-Based | Fitting Error |
| Jung [28] | -Filtering -EC | -Voxel | × | -Straight line on X-Y plane polynomial on Y-Z plane | -Numerical method | 213.6 | -C$_p$ 93.9 -C$_r$ 96.65 | NA | 0.007 |
| Lehtomaki [58] | -RANSAC | -2D plane | × | NA | -NA | 24 | -C$_p$ 93.6 -C$_r$ 93.9 | NA | NA |
| Zhenwi [59] | -EC -PL direction | -NA | × | -Catenary | -Numerical method | NA | -C$_p$ 98.06 -C$_r$ 99.8 | NA | NA |
| Fan [60] | -EC -Segment matching | NA | ✓ | -Straight line on X-Y plane polynomial on Y-Z plane | -LS | NA | -C$_p$ 10 -C$_r$ 98 | NA | 0.015 |
| Gaha [61] | -RANSAC | NA | ✓ | -Straight line on X-Y plane polynomial on Y-Z plane | -Numerical method | NA | NA | -C$_p$ NA -C$_r$ 98.9 | NA |
| Guan [22] | -HT -EC | NA | ✓ | -Polynomial | -Numerical method | 286.4 | NA | -C$_p$ 98.84 -C$_r$ 99 | 0.06 |
| Yadav [19] | -HT -EC | 2D plane | ✓ | -Parabola | -LS | 16.70 | NA | -C$_p$ 90.84 -C$_r$ 98.84 | NA |

Only a few methods have been able to estimate the root means square error (RMSE). The overall comparison shows that the methods based on span locations for the extraction and reconstruction comparatively showed better completeness and correctness [51,70].

The Guo et al. [71] method based on segmentation achieved 90% completeness while the Zhang [29] method based on clustering achieved 100% completeness. However, Guo et al. [71] claimed that their method can reconstruct power lines with a gap of 10 m, whereas the Zhang [29] method will be affected if the gap is greater than 4 m, and the Zhou [74] method would fail if the gap is greater than 6 m. Munir et al. [26] and Zhou et al. [74] were able to reconstruct four sub-conductors with a precision of 0.2 m, respectively, while Awrangjeb [27] was able to reconstruct two sub-conductors with a modelling error of 0.001 m.

Until now, only a few studies have focused exclusively on the extraction of power lines from MLS point clouds. Mostly, these studies have concentrated on urban environments and extracted single power line spans, not only due to the obvious mobility limitations of a standard MLS mounted on a car, but also due to the technology's relatively recent emergence.

## 4. Discussion

### 4.1. Summary of Methods for Power Lines Extraction and Reconstruction

Based on our examination in Section 3, we may assume that, when LiDAR technology was originally released, the classification of power line scenes was the primary focus. The classification findings obtained using machine learning techniques are more applicable and accurate than rule-based classification. Furthermore, some studies trained the data with numerous classifiers and neighbourhoods, resulting in higher precisions and accuracies [48]. Using post-processing procedures can help improve classification [56]. On the other hand, the rule-based classification does not require training data, but methods rely on a number of parameter settings.

As LiDAR technology matured, utility companies looked for improved PLC survey alternatives. Power line modelling calls for conductor-specific points to simulate 3D power lines. Numerous studies have examined the extraction of a single power line using fixed-wing and rotor-wing aircraft.

The majority of methods employ some common steps, i.e., classification of power line scene first using machine learning classifiers [74] or pre-processing steps to remove ground and other non-power line points using various filtering techniques such as density and height filtering [28]. To find individual power line points, many studies used Euclidean clustering algorithms as well [76]. In addition, some methods use the HT algorithm [75] to detect power lines and then apply clustering to connect the power line segments.

As research increased, the point cloud grew in size, and managing the big file size became a new difficulty. In numerous studies, a voxel grid was utilised to address this issue by transforming the irregular structure of point clouds into regularly spaced sub-sampled points or filtering features to preserve the essential details contained in the point cloud while drastically reducing the dataset size [28,51]. Some approaches have also showed that the location of pylons can facilitate the extraction of power lines in the form of spans [76]. As the configuration of power lines remains the same in each span, extracting the spans first will aid in extracting the bundle conductors and modelling the power lines with the same parameters. This will reduce the size of the point cloud, making it easier to isolate the power lines from each span [27]. Additionally, it minimises the effort required to process LiDAR data.

For reconstruction of power lines, the studies given in literature are grouped into two categories. The first category of studies used the idea of extracting the un-organised candidate power line points and then employed clustering algorithms to group power line points into clusters. Mathematical models were then fitted to seed clusters, which were grown by adding adjacent points consistent with the trend of the model [21,28,69].

However, in other categories, the mechanism is developed using image-and point-based methods to extract the segments of power lines and then the segments are modelled using the mathematical models such as parabola [19], catenary [69,71,79,80], and these segments are considered as seed regions and extended by estimating the parameters using the algorithms such as the RANSAC and least square algorithms [71,74].

High voltage wires in bundle configuration as shown in Figure 4 are required to meet the increased demand of electricity and to transfer electricity city-to-city. In order to perform maintenance of high voltage PLC, sub-conductor modelling is required which subsequently required sub-conductors points extraction from the bundle.

As discussed in Section 3.2 and depicted in Figure 4, extracting the sub-conductor from the bundle is not a simple task. With a distance of 0.25 metres, the sub-conductors from the bundle are located very close together. Their separation is extremely difficult. Awrangjeb [27] first proposed the concept of sub-conductor extraction and was able to extract bundles with two sub-conductors in each. Only a few studies in literature have focused on this problem, and among these these studies, Munir et al. [26,51,75,76] was capable of extracting and reconstructing up to our sub-conductors in each bundle with high precision and accuracy.

MLS/TLS data have been utilised in numerous power line extraction studies in recent years. These methods provide techniques for the extraction of a single power line but make no provisions for the extraction of multiple conductors. As an MLS system generates extremely dense point clouds, they can be combined with multi-conductor extraction techniques to detect buildings and other undesirable objects. Most studies employed the filtering algorithm to refine numerous points [19,22,60] and to extract candidate power line points by means of segmentation [28]. Then, clusters are filtered using RANSAC [61] or PCA [58,59].

The testing area of these methods was mostly on urban areas [22] or rural areas near road way environments [19], due to mobility issues of MLS. However, there are few studies [28,61] that have tested their methods on forest locales as well and achieve good accuracies. As MLS is a relatively new technology, the datasets have high densities, which is very important for the accurate extraction of power lines.

Due to the variety of study topics and evaluation methods, it is impossible to compare accuracies of the available methods for power line extraction and reconstruction.

### 4.2. Topics for Further Research

PLC modelling, particularly the automated extraction of power lines, is one of the most commonly explored applications of LiDAR data. In terms of overall feasibility for reliable automatic detection and detailed 3D reconstruction of conductors and poles, ALS and MLS point clouds appear to be the most promising technologies. The methodologies and expected level of information in the results vary substantially depending on the datasets and the quality of the LiDAR data. However, due to the diversity of study topics and quality assessment methodologies, it was hard to draw broad judgements about the amount of accuracy that could be attained or to compare different approaches. Findings of many studies have merely been provided in a few figures. Larger real-world tests, as well as numerical quality analyses, appear to be unusual. This, in conjunction with the vast amount of continuing research, shows that even simple problems, such as automatic recognition of power line conductors from point clouds, have yet to be fully solved and tested. Comparing the available options to assess their applicability in real time required the usage of common data sets and evaluation tools. As demonstrated in the tables, some approaches used point-based quality assessments, while others used object-based quality measurements.

In recent years, feature learning approaches have been adapted to address point clouds [81–83], inspired by dense convolution, which can acquire translation in variance. The possibility of applying deep learning (DL) to ALS point cloud for power line scene classification using the graph data structure was recently investigated in [84] with some initial results. DL approaches for categorising power line scenes are currently in their infancy and require further development.

Multi-source data, on the other hand, are rarely used in research. The following are some possible future research topics in this area: Optimal integration of ALS and MLS data for detailed but cost-effective 3D mapping of power lines. To achieve versatile and detailed monitoring capability, standard remote sensing techniques are combined with climbing

robots and camera systems on power line structures. Remote sensing techniques such as UAVs and laser scanning from airborne and land-based platforms should be given special attention in any future development.

Typically, the starting point for power line extraction in the existing literature is typically remotely sensed data from a single date, with no prior knowledge of the power line structures' location. However, some map data are available in many cases, and the map data could be used as a starting point for change detection or as a guide for the detection and modelling process. This is a topic that should be investigated in order to maximise the use of existing data and possibly achieve a higher level of automation. This is also critical in terms of map data updates. Furthermore, existing map data can be used to guide UAVs flying around power lines.

Change detection based on remote sensing, in which datasets from two or more dates are compared to each other, is another topic worth investigating in the context of power line monitoring. This could be a good way to detect network component problems. Remote sensing techniques such as UAVs and laser scanning from airborne and land-based platforms, which are rapidly developing, should be given special attention. These techniques will almost certainly be used to develop effective, flexible, and highly automated monitoring methods, allowing commercial applications.

Large test datasets in various environments and realistic monitoring conditions are required to demonstrate and verify the capabilities of automated monitoring approaches. These should also cover difficult situations such as power lines in the forests. Careful quality analyses and comparisons between different data sources, methods, and individual algorithms are required to develop efficient integrated approaches.

## 5. Conclusions

LiDAR surveying is preferred for regular power line corridor (PLC) inspection. The literature contains many state-of-the-art laser scanning-based techniques for the extraction and reconstruction of overhead power lines. Laser scanning techniques are commonly used in ALS (fixed-wing aircraft, helicopters, unmanned aerial vehicles (UAVs), and terrain-based (mobile laser Scanning (MLS)/terrestrial laser scanning (TLS)) approaches. ALS is commonly used for power line detection due to its large scanning range, but its low accuracy is due to the long distance between the destination and its scanner. MLS, on the other hand, scans closer to targets than ALS, allowing for more precise data collection.

Numerous publications have focused on classification and reconstruction techniques for power line extraction, specifically conductor extraction. A typical procedure includes creating a DTM, classifying the laser data to distinguish power line points from other objects, particularly vegetation, and modelling the individual conductors in 3D. Previously, the emphasis was on classifying power line points using supervised and unsupervised ALS methods, but there has been a growing interest in recent years in examining the extraction and reconstruction of power line spans with single and multi conductor configurations using various image and point-based techniques.

This research looked at the technical and methodological limitations and provided a comprehensive overview of the methodologies offered by various approaches using laser scanning data from the perspective of power line extraction applications, as well as discussed the benefits and drawbacks of each approach. This article summarised the findings of existing works, and major issues are highlighted for power line extraction and reconstruction using LiDAR point cloud data.

Despite the tremendous potential of aerial and mobile scanning systems, the comparison revealed that human intervention and post-processing actions are still required to achieve the desired results. Furthermore, the majority of the methods were evaluated on small datasets on spans rather than large datasets, and only a few methods were focused on multi-conductor extraction and reconstruction for power line modelling. These impediments impede the automated extraction and reconstruction of power lines from LiDAR data. Several promising directions for future LiDAR experiments using deep learning

methods are outlined in the hope that they will pave the way for PLC modelling and assessment applications at a finer and larger scale.

**Author Contributions:** Conceptualization, N.M. and M.A.; methodology, N.M. and M.A; software, N.M.; validation, N.M. and M.A.; formal analysis, M.A.; investigation, N.M.; resources, N.M.; data curation, M.A.; writing—original draft preparation, N.M.; writing—review and editing, M.A.; visualization, M.A.; supervision, M.A. and B.S. All authors have read and agreed to the published version of the manuscript.

**Funding:** This research received no external funding.

**Data Availability Statement:** Datasets are unavailable due to privacy or ethical restrictions.

**Acknowledgments:** The authors would like to express their gratitude to the editors and the reviewers for their constructive and helpful comments for the substantial improvement of this paper.

**Conflicts of Interest:** The authors declare no conflict of interest.

## Abbreviations

The following abbreviations are used in this manuscript:

| | |
|---|---|
| PLC | Power line corridor |
| CLF | Campus Line Filter |
| DL | Deep Learning |

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
