# Peer review of "Power Line Extraction and Reconstruction Methods from Laser Scanning Data: A Literature Review"

_remotesensing, doi:10.3390/rs15040973_

Round 1

Reviewer 1 Report

This paper summarizes power line extraction and reconstruction methods based on laser scanning, and provides useful guidance for researchers using innovative LIDAR scanning systems for power line 3D modeling. The language of the paper is clear and can clearly express the relevant progress in the field. I think this article can be accepted after a minor revision. The following comments are for the author's reference.

L60: Can the author give some methods based on remote sensing in detail? For example, in satellite-based remote sensing monitoring, which data are available, how do they work?

L99: The author mentions nearly 50 studies. Which 50 studies? Perhaps the author can provide an attachment.

L166: I don't know why 3.1.1 is used, and there is no section 3.1.2 in the text below the article, so is it necessary? Perhaps the previous content is section 3.1.1.

L270: Is there a more detailed flow chart of the process? A flow chart should preferably show the relationship between the columns of Tables 2-5.

L294: Maybe you should use three levels of headings.

Table 2: Zhou [], Missing reference.

L320: see column 32? What does that mean?

L676: The article only has section 5.1, not section 5.2.

Finally, the format of the article needs a good revision.

Reviewer 2 Report

This is a great article that reviews the powerline extraction using laser scanning technology.  I enjoyed reading it. A couple of minor edits are needed:

1. I suggested moving Pg 2, contribution to the end of the manuscript

2. Page 21, Line 792-806, Please do not use the template and provide your own information.

Reviewer 3 Report

The manuscript summarizes and presents several very interesting issues about power line extraction and modeling. They are very revealing.

However, it has some questions that need to be carefully revised and added.

in Line 707, the last sentence is used to prove what problem?

Please correct some errors in the text (see attachment), lines 556,328,713,520...

The last paragraph of the manusctript is repetitive and should be deleted, and the paragraph (line 709) should be merged with the paragraph (line 715); the paragraph (line 95) needs to be simplified.

Figure 2 is borrowed from other articles with only minor modifications and it is suggested to include citations.

This article aims to analyze the progress and problems of power line extraction, so it is not necessary to discuss for vegetation monitoring in the corridor, in chapter 5; in addition, the writing of this chapter is not well organized, and finally, for ' limitations that raise a general scepticism for power line mapping techniques using laser scanning data' it is suggested to explain in detail.

Round 2

Reviewer 3 Report

no commets